# A Highly Selective and Sensitive Nano-Silver sol Sensor for Hg^2+^ and Fe^3+^: Green Preparation and Mechanism

**DOI:** 10.3390/polym14183745

**Published:** 2022-09-07

**Authors:** Yining Yang, Xiaodong Zhou, Ruitao Dong, Yanwei Wang, Zichao Li, Yun Xue, Qun Li

**Affiliations:** 1College of Chemistry and Chemical Engineering, Shandong Collaborative Innovation Center of Marine Biobased Fibers and Ecological Textiles, Qingdao University, Qingdao 266071, China; 2Institute of Biomedical Engineering, College of Life Sciences, Qingdao University, Qingdao 266071, China

**Keywords:** colorimetric chemosensor, silver nanoparticles, Hg^2+^, Fe^3+^, green preparation

## Abstract

The development of highly selective and highly sensitive nanometer colorimetric chemical sensors is an urgent requirement in the immediate detection of heavy metal ions. In this work, silver-nanoparticle (Ag NPs)-based chemosensors were prepared by a simple and green method, in which the silver nitrate, carboxymethyl cellulose sodium (CMS) and Polyvinylpyrrolidone (PVP), and glucose are used as the silver source, double stabilizer and green reductant, respectively. The obtained colloidal CMS/PVP-Ag NPs showed a high dispersibility and stability, and creating a high selectivity and sensitivity to detect Hg^2+^ and Fe^3+^ with remarkable and rapid color variation. Low limits of detection (LOD) of 7.1 nM (0–20 μM) and 15.2 nM (20–100 μM) for Hg^2+^ and 3.6 nM for Fe^3+^ were achieved. More importantly, the CMS/PVP-Ag NPs has a high sensitivity even in a complex system with multiple heavy ions, the result of the practical ability to detect Hg^2+^ and Fe^3+^ in tap water and seawater reached a rational range of 98.33~104.2% (Hg^2+^) and 98.85~104.80% (Fe^3+^), indicating the great potential of the as-prepared nanocomposites colorimetric chemosensor for practical applications.

## 1. Introduction

Facing the hazardous problem of heavy metal pollution in the air, water and even food, it is imperative to improve detection methods [1,2,3]. Among the various heavy metal ions, excess intake of foods containing divalent mercury (Hg^2+^) and ferric iron (Fe^3+^) in daily life can harm the human body, resulting in brain damage, Alzheimer’s disease, Parkinson’s disease and endocrine system disease [4,5,6,7,8,9]. Thus, developing analytical detection techniques for Hg^2+^ and Fe^3+^ has emerged as a significant research field [10,11,12,13,14]. Several conventional analytical technologies, such as atomic absorption spectrometry (AAS), inductively coupled plasma mass spectroscopy (ICP-MS) and high-phase liquid chromatography (HPLC), have been widely applied due to the excellent sensitivity and selectivity for the detection of heavy metal ions [15,16,17,18]. Unfortunately, the fact that those technologies involve high cost, are time consuming and have complicated operations has greatly affected their extensive application in the detection of metal ions [19,20,21].

Currently, colorimetric chemosensors have the prominent advantage of high sensitivity and being visible to the naked eye [22,23,24]. Intriguingly, benefiting from the optical, mechanical and electrochemical properties, the NPs manifest diverse applications in different fields (energy storage, catalysis, coating and absorption) [25,26]. To date, a series of metal NPs-based colorimetric sensors have been developed owing to the outstanding extinction coefficients in the visible area of colorimetric assays. Silver nanoparticles (Ag NPs) can provide higher extinction coefficients than gold nanoparticles (AuNPs) [27,28] and have a plasma resonance absorption (SPR) [29,30,31], and the sensitive central absorption peak of Ag NPs is located at 400 nm [32], suggesting the desirable applicability of colorimetric sensing of metal ions [33], which are attributed to the good interaction between Ag NPs and its analytes, thus changing the position of the absorption peak in the spectrum [34,35]. The construction of functional Ag NPs is essential in determining the properties of colorimetric sensing as a green-reducing agent [36], and the introduction of glucose offers the aldehyde groups to effectively reduce the silver nitrate to Ag NPs [37,38]. The above sensors have the defects of complex synthetic conditions, the release of harmful synthetic raw materials into the environment and a high LOD. Therefore, the development of a green, rapid, environmentally friendly and low-LOD detection method based on nano silver sol colorimetric chemical sensors has great potential application prospects.

Here, we fabricated an efficient colorimetric chemosensor (CMS/PVP-Ag NPs colloid) via a facile and green strategy. Using CMS and PVP as double green stabilizers, CMS/PVP-Ag NPs were prepared by a silver mirror reaction of glucose and AgNO_3_ and the Ag NPs were anchored on the stable CMS/PVP network via non-covalent bonds. As a result, the obtained CMS/PVP-Ag NPs colloid showed high dispersibility and stability and exhibited excellent sensitivity for Hg^2+^ and Fe^3+^ with a reasonable LOD value, selectivity and, in particular, the colorimetric chemosensors had superior selectivity in multiple ion systems. Of note, the colorimetric chemosensors could effectively probe the content of Hg^2+^ and Fe^3+^ in tap water and seawater, suggesting great potential applications of detection within complex water samples.

## 2. Experimental Procedures

### 2.1. Reagents and Apparatus

AgNO_3_, Glucose (Glu), Carboxymethyl cellulose sodium (CMS), Polyvinylpyrrolidone (K30, PVP), Al(NO_3_)_3_·9H_2_O, Cu(NO_3_)_2_·3H_2_O, Bi(NO_3_)_3_·5H_2_O, Ba(NO_3_)_2_, Ca(NO_3_)_2_·4H_2_O, Cd(NO_3_)_2_·4H_2_O, Co(NO_3_)_2_·6H_2_O, Fe(NO_3_)_2_·6H_2_O, Fe(NO_3_)_3_·9H_2_O, KNO_3_, Mg(NO_3_)_2_·6H_2_O, Ni(NO_3_)_2_·6H_2_O, Pb(NO_3_)_2_, Zn(NO_3_)_2_·6H_2_O, HgCl_2_ and NaNO_3_ reached analytical-reagent grade and were purchased from Sinopharm Chemical Reagent Co., LN (Shanghai, China).

### 2.2. Synthesis of CMS/PVP-Ag NPs Colloidal Solution

The detailed synthesized process of the CMS/PVP-Ag NPs colloid is illustrated in Figure 1. At first, 50 μL of 0.1 M AgNO_3_ was added into CMS (0.1% [*w/v*], 50 mL) and PVP (1% [*w/v*], 5 mL) aqueous solution with vigorous magnetic stirring for 10 min. After that, glucose (0.1 M, 150 μL) and NaOH (1% [*w/v*], 800 μL) were transferred into the above mixture solution with continuous stirring for 1 h. The reaction was finished until no color changes and stored in dark conditions. All of the experimental processes were performed at a constant temperature of 70 °C.

### 2.3. Characterization of CMS/PVP-Ag NPs Colloidal Solution

The UV–VIS absorption spectra were recorded via UV–VIS spectrophotometer (UV-2450, Shimadzu, Kyoto, Japan) with a variable wavelength between 300 and 600 nm at room temperature. X-ray diffractometer (XRD) data were measured by a powder X-ray diffractometer (D/MAX-RB, Tokyo, Japan) using CuKα radiation (λ = 0.15418 nm) over the 2θ range of 30° to 80° with a step-size of 0.05°. Transmission electron microscopy (TEM) images were obtained using a JEOL JEM-2100 Plus microscope (JEOL, Tokyo, Japan) at an accelerating voltage of 200.0 kV.

### 2.4. Adsorption and Detection of Hg^2+^ and Fe^3+^

The CMS/PVP-Ag NPs colloidal solution was diluted twice with ultra-pure water, and then utilized for the adsorption and detection of Hg^2+^ and Fe^3+^. In order to investigate the adsorption and detection properties of the CMS/PVP-Ag NPs colloidal solution, various metal ions (60 µL, 0.001 M) including Mg^2+^, Co^2+^, Cu^2+^, Ni^2+^, Zn^2+^, Ca^2+^, Cd^2+^, Fe^3+^, Fe^2+^, K^+^, Ba^2+^, Na^+^, Hg^2+^, Al^3+^ and Pb^2+^ ions were added to the CMS/PVP-Ag NPs colloidal solution (3 mL), respectively. Then, the above solutions were shaken well to observe color changes with the naked eye and were measured by the UV-vis spectrophotometer. Compared with the blank solution, it was found that the Ag NPs solution with Hg^2+^ and Fe^3+^ had apparent changes in color and ultraviolet absorption. To further obtain the detection limit, different concentrations of Hg^2+^ ions (3–300 µL,1 mM) were added to the CMS/PVP-Ag NPs colloidal solution (3 mL) and the quantitative detection of Fe^3+^ (3–30 µL, 1 mM) was also completed according to the above procedure. Additionally, samples were collected from seawater and animal plasma to confirm the practical application of the CMS/PVP-Ag NPs colloidal solution.

The limit of detection (LOD) is the lowest concentration of the measured sample, and the calculation formula is:(1)LOD=3S/N
where S represents the standard deviation of the blank sample, and N is the slope of the standard linear.

## 3. Results and Discussion

### 3.1. Characterization Studies of CMS/PVP-Ag NPs Colloidal Solution

The synthesis method of a silver-nanoparticle (Ag NPs)-based chemosensor used CMS as the stabilizing agent and glucose as the reducing agent [39]. The Ag NPs sol was synthesized by adding AgNO_3_ solutions of different concentrations and the obvious characterization adsorption was presented in the region of 410–430 nm. As shown in Figure 1a, the absorbance intensity of the Ag NPs sol prepared increased with the increase in AgNO_3_ concentration, and there was a small blue shift, indicating that sol aggregation occurred easily at high concentrations. Figure 1b shows the UV-vis spectra of silver sol prepared by AgNO_3_ and glucose with different molar ratios. From the absorption curve, there was no significant difference between the molar ratios 1:2 and 1:3. However, after a week, the absorption curve of AgNO_3_/glucose = 1/3 almost changed (Figure 1c), indicating that the CMS/PVP-Ag NPs colloid solution possessed the best stability.

Figure 2a shows the XRD pattern of Ag NPs to analyze the crystalline phase. It can be seen that there was an obvious crystal diffraction peak at 2θ = 38.5°, which can be assigned to the (111) of silver crystal according to the JCPDS card. In addition, there are three weak peaks at 2θ of 44.5°, 64.5° and 77.6°, corresponding to the (200), (220) and (311) crystal planes of the crystalline silver, respectively, indicating the formation of simple silver in the system [40,41]. The morphology of Ag NPs was characterized by the TEM image (Figure 2b). Clearly, Ag NPs exhibit well-defined spherical crystals, and the SAED pattern of CMS/Ag NPs shows bright circular rings, proving the polycrystalline characteristic of the synthesized Ag NPs [42]. These particle diameters are distributed in the range of 15 to 40 nm and the calculated average nanocrystal size of Ag NPs was approximately 25 nm (Figure 2c). Meanwhile, the EDS results suggest the distribution of the Ag element (Figure 2d) [43].

### 3.2. Detection Selectivity and Sensitivity of Hg^2+^ and Fe^3+^

The various metal ions were selected (such as Hg^2+^, Cd^2+^, Co^2+^, Cu^2+^, Fe^3+^, Fe^2^^+^, K^+^, Al^3+^, Ba^2+^, Ca^2+^, Mg^2+^, Na^+^, Ni^2+^, Pb^2+^ and Zn^2+^) to verify the selectivity of the CMS/PVP-Ag NPs colloidal solution. As shown in Figure 3a, all ions were treated as detected under the same conditions with a maximum concentration of 50 μM, and only Hg^2+^ and Fe^3+^ produced noticeable response curves and the absorption peak decreased with an obvious blue shift after the addition of Hg^2+^ and Fe^3+^. Moreover, the values of ΔA were found to be 0.62 and 0.3 in the presence of Hg^2+^ and Fe^3+^, respectively, and the obvious changes could not be found in other metal ions (Figure 3b). When Hg^2+^ and Fe^3+^ are added to 50 μM, the color of the CMS/PVP-Ag NPs colloidal solution tended to be colorless and lighter, respectively, further demonstrating that the CMS/PVP-Ag NPs colloidal solution has a unique selectivity for Hg^2+^ and Fe^3+^ rather than other metal ions [44].

Moreover, the sensitivity of the CMS/PVP-Ag NPs colloidal solution for Hg^2+^ and Fe^3+^ was further investigated to probe the optimum conditions. It was shown that the color of the sensing system gradually faded away with increasing concentrations of Hg^2+^, and when the concentration of Hg^2+^ increased to 50 μM, the color of the sensing system disappeared, providing an efficient platform for the colorimetric detection of Hg^2+^ (inset in Figure 4a). On the addition of Hg^2+^, the absorption peak of Ag NPs decreased and the maximum absorption wavelength of Ag NPs displayed a distinct blue shift (Figure 4a), which indicated the formation of a Ag–Hg amalgam on the surface of Ag NPs and the decrease in the concentration and size of the nano-silver particles [45]. The correlation of the ion concentration in the range of 0 to 100 μM and the absorption response were exhibited in Figure 4b; there were two good linear relationships (R_1_ = 0.998, R_2_ = 0.995) between ΔA and Hg^2+^ (0–100 μM), and the LOD for Hg^2+^ was estimated to be 7.1 nM (0–20 μM) and 15.2 nM (20–100 μM). Meanwhile, Figure 4c reveals two favorable linear relationships (R_1_ = 0.980, R_2_ = 0.991) between Δλ and the Hg^2+^ concentration in the range of 0 to 100 μM, and the LOD was approximately 0.2 nM (0–30 μM) and 0.7 nM (30–100 μM), respectively. The value of LOD, due to the addition of the small amount of Hg^2+^, could significantly decline when the absorbance was added, causing the higher sensitivity of detection, which is comparable with or superior to other detection methods in previous literature (Appendix A) [46,47,48,49,50,51]. The excellent LOD value with small Hg^2+^ concentrations and the wide linear range implied a high sensitivity and potential application for Hg^2+^ detection.

As shown in Figure 5a, a slight decrease in the absorbance of Ag NPs could be observed with a gradual increase in the concentration of Fe^3+^. It is noted that the ΔA and the concentration of Fe^3+^ in the range of 0–60 μM maintained a good linear relationship (R = 0.993) and the value of LOD was approximately 3.6 nM, demonstrating that the detection ability of the system was relatively significant (Figure 5b). Appendix A shows a comparison of our sensors with those previously reported in literature [5,13,15,17,19,52,53,54,55]. The method is green, rapid, simple, low-cost and highly sensitive. More importantly, the as-prepared sensors showed good stability, and were highly sensitive and selective with a low LOD and the wide linear range for the detection of Fe^3+^.

The TEM image indicates the aggregation of Ag NPs after the addition of Hg^2+^, demonstrating that the addition of Hg^2+^ can induce the formation of a Ag–Hg amalgam (Figure 6b) [44]. Figure 6c reveals that the size of Ag NPs is estimated to be 50–150 nm, which corresponds with the TEM image. Additionally, the result of EDS shows three peaks of Hg (Figure 6d), which is consistent with the above results.

TEM was conducted to explore the morphology variation of the CMS/ PVP-Ag NPs sol when the Hg^2+^ was introduced. Figure 7a presents the image of CMS/PVP-Ag NPs with Hg^2+^, and Figure 7b and c reveals the existence of mercury and silver elements in the CMS/PVP-Ag NPs, respectively. A superimposed image of mercury and silver are shown in Figure 7d. It can be observed that the composites are hybridizing with each other, and adding Hg^2+^ into the CMS/PVP-Ag NPs can increase the size of Ag NPs to form a large cluster.

### 3.3. Mechanism Study for the CMS/PVP-Ag NPs Colloidal Solution and the Detection of Hg^2+^ and Fe^3+^

Figure 8 exhibits the reaction scheme and mechanism of the CMS/PVP-Ag NPs colloid for the detection of Hg^2+^ and Fe^3+^. The synthesis processes involve adsorption, in situ reduction and a complexation/oxidation process. As previously mentioned, CMS and PVP were self-assembled to form the stabilized skeleton and the silver ions were uniformly anchored on the surface of the CMS molecules via a non-covalent bond. Subsequently, the glucose molecules containing aldehyde groups act as the reducing agent to form the Ag NPs through a silver mirror reaction. The larger steric hindrance effect caused the Ag NPs to be uniformly dispersed and not easy to coalesce, as shown in Figure 8b. It was observed that when Hg^2+^ was added into the CMS/PVP-Ag NPs colloidal solution, it became colorless and the SPR band intensity diminished with a blue shift (see Figure 4a), which can also be demonstrated by the morphology changes shown in Figure 8a. The size of Ag NPs was changed from about 25 nm to 50–150 nm in Figure 8b and induced the aggregation of Ag NPs. Furthermore, the detective absorption of Hg^2+^ was examined by a redox reaction with Ag^0^ and Hg^2+^. The generated Hg^0^ adhered to the surface of Ag NPs, forming the Ag–Hg amalgam to enlarge the aggregation of Ag NPs (Figure 8b) [45]. When Fe^3+^ was added to the CMS/PVP-Ag NPs colloidal solution, it became colorless and the SPR band intensity diminished without any red or blue shift in Figure 5a. It indicated that the concentration of Ag NPs in the solution decreases, but the size of Ag NPs remained unchanged, which can be attributed to the oxidation of the Ag^0^, leading to the decomposition of Ag NPs, reducing the SPR band strength and making the solution colorless (Figure 8b) [56,57].

### 3.4. Detection of Hg^2+^ and Fe^3+^ in Tap Water and Seawater 

To test the feasibility and practicability of composites, the prepared product was applied to monitor tap water and seawater. The tap water and seawater were, respectively, taken from our laboratory and China’s Yellow Sea. Firstly, the tested water samples were filtered with a 0.22 μm membrane to remove insoluble matter. Secondly, three known concentrations of Hg^2+^ and Fe^3+^ were added to CMS/PVP-Ag NPs colloidal solution with constant stirring, maintained for 1 min and we observed the color change with the naked eye. Lastly, the water samples with different concentrations of ions were measured with UV-vis spectrophotometer and the recoveries of Hg^2+^ and Fe^3+^ were calculated in real samples. We measured the recoveries of Hg^2+^ in real samples with a range of 98.33~104.2% (Table 1) and the recoveries of Fe^3+^ in water samples ranged from 98.85% to 104.80% (Table 2). The results show that this method can provide accurate and reliable results for the determination of Hg^2+^ and Fe^3+^ in tap water and seawater. Our colloidal solution could be used to detect Hg^2+^ and Fe^3+^ in water samples for practical applications.

## 4. Conclusions

In summary, we have developed an efficient and effective colorimetric chemosensor CMS/PVP-Ag NPs colloid through a facile strategy, which had superior detection selectivity for Hg^2+^ and Fe^3+^. At the addition of Hg^2+^, Ag NPs nanoparticles were induced to enlarge the size of the nanoparticles. Benefiting from the existence of an obvious color change for metal ions and localized SPR absorption of Ag NPs, the colloidal solution exhibited prominent colorimetric responses from yellow to white in the presence of Hg^2+^ and from yellow to buff in the presence of Fe^3+^; the achromatic mechanism may be that Ag–Hg alloy was formed by an oxidation-reduction reaction between Hg^2+^ and nano silver ions, and Fe^2+^ was formed by Fe^3+^. CMS/PVP-Ag NPs exhibited excellent selectivity for Hg^2+^ and Fe^3+^ with low limits of detection of 1.4 μg L^−1^ (7.1 nM) and 0.2 μg L^−1^ (3.6 nM), respectively. Additionally, a simple method for detecting the concentration of Hg^2+^ and Fe^3+^ in tap water and seawater was used, proving their bright prospects to be practically applied in real analysis.

## Data Availability

The data presented in this study are available on request from the corresponding author.

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
