# Peer review of "A Highly Selective and Sensitive Nano-Silver sol Sensor for Hg2+ and Fe3+: Green Preparation and Mechanism"

_polymers, 2022, doi:10.3390/polym14183745_

Round 1

Reviewer 1 Report

EXCELLENT WORK

I HIGHLY RECOMMEND ITS PUBLICATION AS IT IS WITHOUT CORRECTIONS

Author Response

Thank you very much for the recommendation.

Reviewer 2 Report

Dear Editor

The paper describes the detection of Hg2+ and Fe3+  using Nano-silver sol Sensor. There is a novelty in this work. Authors have used Nano-silver sol Sensor. The paper provides relevant data to the study of detection of Hg2+ and Fe3+, however at this stage the text has misleading information, and has several errors (tipos) that have to be corrected. Some of the interpretations made need further support data or have to be corrected. The manuscript seems to be suitable for publishing if revised (major revision) according to the comments presented below.

1-      Why the authors used colorimetric chemical sensors?

2-      The authors need to talk about thermal stability of silver nanoparticle (Ag NPs)

3-      The author needs to explain why used this type method and why silver nanoparticle (Ag NPs).

4-      The author needs to explain why choose Hg2+ and Fe3+   

5-      Should be a single space between the number and unite.

6-      In the literature review, the author should be mention  the following work:

·         Voltammetric Determination of Hg2+, Zn2+, and Pb2+ Ions Using a PEDOT/NTA-Modified Electrode..  https://pubs.acs.org/doi/full/10.1021/acsomega.2c02682, https://doi.org/10.1021/acsomega.2c02682

·         Use of a Schiff base-modified conducting polymer electrode for electrochemical assay of Cd (II) and Pb (II) ions by square wave voltammetry.. https://link.springer.com/article/10.1007/s11696-021-01882-7 , https://doi.org/10.1007/s11696-021-01882-7

7-       In  conclusion, the author should be clearly mentioned what are the best conditions for detection  and why.

8-       There are many grammar and spilling mistakes.

Author Response

Thank you very much for the recommendation, and the constructive and invaluable comments. According to the following specific comments, we have carefully revised and improved the manuscript to merit its publication in Polymers.

Reviewer 3 Report

This manuscript provides a colorimetric sensor for Hg2+ and Fe3+detection. The author claims they have developed a highly selective and sensitive nano-silver sol sensor for Hg2+ and Fe3+ in green preparation and mechanism manner. However, in my opinion, the mechanism reported there is really similar to some previous work, such as Journal of Molecular Liquids, 2020, 307, 112978. Journal of Molecular Liquids, 2020, 311, 113281, Journal of Materials Chemistry, 2011, 21, 5190, etc. In addition, the performance of the sensor is also not such well compared to the literature reported (for example, the sensor does not seem to be sensitive to the detection of Fe3+, even when the concentration of Fe3+ reaches 50 μM from Figure 3; and the specificity of the sensor is also not high, since it can respond to both Hg2+ and Fe3+, etc.). I therefore do not think this work can be published in this journal at present.

Author Response

Thanks for giving the precious comment. In previous work, AgNPs were typically synthesized by NaBH4 to reduce the AgNO3 in ice chilled condition. In this work, we reported a green and simple synthesized strategy of Ag NPs by using CMS as the stabilizing agent and glucose as the reducing agent. The larger steric hindrance effect made that the Ag NPs were uniformly dispersed and not easy to coalesce. Additionally, the prepared CMS/PVP-Ag NPs can simultaneously detect Hg2+ and Fe3+ and exhibit low limits of detection of 7.1 nM (1.4 μg L-1) and 3.6 nM (0.2 μg L-1), respectively. Table S1 and S2 show a comparison of Hg2+ and Fe3+ detection with those reported literature, demonstrating that our sensor has potential application for Hg2+ detection.

Round 2

Reviewer 2 Report

Accept in present form

Reviewer 3 Report

Generally, I think the authors are perfunctory to my previous comments. I do not think they really take care of some questions. Herein, I will reiterate my concern, “The performance of the sensor is also not such well compared to the literature reported (for example, the sensor does not seem to be sensitive to the detection of Fe3+, even when the concentration of Fe3+ reaches 50 μM from Figure 3; and the specificity of the sensor is also not high, since it can respond to both Hg2+ and Fe3+, etc.).”  Please be noted that low LOD does not means everything. 

Therefore, I do not think I will recomend this work for publication at present, sorry.